# Randomized Controlled Trial of Two Timepoints for Introduction of Standardized Complementary Food in Preterm Infants

**DOI:** 10.3390/nu14030697

**Published:** 2022-02-07

**Authors:** Nadja Haiden, Margarita Thanhaeuser, Fabian Eibensteiner, Mercedes Huber-Dangl, Melanie Gsoellpointner, Robin Ristl, Bettina Kroyer, Sophia Brandstetter, Margit Kornsteiner-Krenn, Christoph Binder, Alexandra Thajer, Bernd Jilma

**Affiliations:** 1Department of Clinical Pharmacology, Medical University of Vienna, 1090 Vienna, Austria; melanie.gsoellpointner@meduniwien.ac.at (M.G.); bernd.jilma@meduniwien.ac.at (B.J.); 2Department of Pediatrics, Medical University of Vienna, 1090 Vienna, Austria; margarita.thanhaeuser@meduniwien.ac.at (M.T.); fabian.eibensteiner@meduniwien.ac.at (F.E.); mercedes.dangl@meduniwien.ac.at (M.H.-D.); sophia.brandstetter@meduniwien.ac.at (S.B.); margit.kornsteiner.krenn@gmail.com (M.K.-K.); christoph.a.binder@meduniwien.ac.at (C.B.); alexandra.thajer@meduniwien.ac.at (A.T.); 3Center for Medical Statistics, Informatics and Intelligent Systems, Medical University of Vienna, 1090 Vienna, Austria; Robin.Ristl@meduniwien.ac.at (R.R.); bettina.kroyer@meduniwien.ac.at (B.K.)

**Keywords:** preterm infant, VLBW infant, complementary feeding, introduction of solids, anthropometry, growth, height, weight, BMI, head circumference

## Abstract

In term infants it is recommended to introduce solids between the 17th and 26th week of life, whereas data for preterm infants are missing. In a prospective, two-arm interventional study we investigated longitudinal growth of VLBW infants after early (10–12th) or late (16–18th) week of life, corrected for term, introduction of standardized complementary food. Primary endpoint was height at one year of age, corrected for term, and secondary endpoints were other anthropometric parameters such as weight, head circumference, BMI, and z-scores. Among 177 infants who underwent randomization, the primary outcome could be assessed in 83 (93%) assigned to the early and 83 (94%) to the late group. Mean birthweight was 941 (SD ± 253) g in the early and 932 (SD ± 256) g in the late group, mean gestational age at birth was 27 + 1/7 weeks in both groups. Height was 74.7 (mean; SD ± 2.7) cm in the early and 74.4 cm (mean; SD ± 2.8; n.s.) cm in the late group at one year of age, corrected for term. There were no differences in anthropometric parameters between the study groups except for a transient effect on weight z-score at 6 months. In preterm infants, starting solids should rather be related to neurological ability than to considerations of nutritional intake and growth.

## 1. Introduction

Adequate nutrition and growth during the first 1000 days, from conception to the second year of life, are crucial for a later healthy life and appropriate neurodevelopment, especially in preterm infants [1]. Along with breast and formula feeding, this period further includes the weaning process and the introduction of solids. In full-term infants, the European Society for Pediatric Gastroenterology, Hepatology and Nutrition (ESPGHAN) [2,3] recommends a stepwise introduction of complementary food between the 17th and 26th week of life. In preterm infants, guidelines on the optimal time for starting solids and the ideal composition of complementary food meeting their special requirements are missing [4,5]. Numerous observational studies documented a wide variability in timing and quality of complementary food, but in general, preterm infants are introduced early to solid foods even when their corrected age is taken into account [1].

So far, only two interventional RCTs on the timing for initiation of complementary food in preterm infants were conducted investigating growth during the first year of life [6,7]. However, these studies were performed under expired settings and different environmental conditions, making a general application of the results impossible. The question of if and how the timepoint of introduction of solids affects the growth of preterm infants during early life remains unclear so far. Therefore, we planned a randomized study in preterm infants with a birthweight below 1500 g, investigating whether the timepoint of introduction of standardized complementary food influences growth at 12 months, corrected for term. 

## 2. Materials and Methods

### 2.1. Study Design

This was a single center, prospective, randomized two-arm interventional trial investigating two different timepoints of introducing a standardized complementary diet in very low birth weight infants. The trial was conducted at the outpatient clinic for preterm infants of the Division of Neonatology, Department of Pediatrics of the Medical University of Vienna, approved by the Ethics committee of the Medical University of Vienna (EK: 1744/2012) and registered on clinicaltrials.gov (NCT01809548).

### 2.2. Participants

Preterm infants with a birthweight below 1500 g, followed up in the outpatient clinic of the Division of Neonatology, Department of Pediatrics, Medical University of Vienna were eligible for the study. 

Exclusion criteria were gastrointestinal diseases (necrotizing enterocolitis stage three [8], Hirschsprung’s disease, chronic inflammatory bowel disease), bronchopulmonary dysplasia [9] defined as oxygen demand after 36 weeks of gestation, congenital heart defects, major congenital birth defects, and chromosomal aberrations. 

### 2.3. Randomization

At term, infants were screened for eligibility and informed consent was obtained from at least one parent (Figure 1). Infants were randomized either to the early complementary food group (introduction of standardized solids between 10th–12th week, corrected for term) or to the late complementary food group (introduction of standardized solids between 16th–18th week, corrected for term, Figure 2). Participants were randomized using permuted blocks (ratio 1:1, block size of six) stratified by breast versus formula/mixed feeding, using the software randomizer (www.randomizer.at (accessed on 28 December 2021)). Twins, triplets, or quadruplets were regarded as one unit in the randomization and assigned to the same group. After an interim analysis in January 2017 and a recruitment status of almost 75% (early complementary food group *n* = 66; late complementary food group *n* = 65), a baseline imbalance in birthweight and gestational age between patients and groups was detected. Infants in the early group had a significantly lower gestational age and a significantly lower birthweight than in the late group. Therefore, the randomization process was switched to a baseline adaptive randomization design in March 2017 with an additional stratification according to birthweight to correct this imbalance.

### 2.4. Procedures and Diet 

During the first year of life, five follow-up visits were scheduled (Figure 2): at term, between 6–8 weeks, between 16–18 weeks, and at six and 12 months, corrected for term. Measured anthropometric data at the respective visits were weight, height, and head circumference. All measurements were performed according to a standard operating procedure (Appendix B). Z-scores for weight, height, head circumference, body mass index (BMI), and weight-for-length were calculated using the WHO-MRGS growth standards [10]. For the calculation of individual growth trajectories, anthropometric data acquired during the hospital stay were included. Growth, height, and head circumference before term were calculated using the Fenton growth charts [11] and included in the anthropometric plots to give additional information on the growth trajectories of the infants during their course in the neonatal intensive care unit up to discharge. The Ponderal Index was calculated as weight in kg/length^3^ in m as previously described [12].

At term infants were breastfed, formula fed, or received mixed feedings on demand. After randomization, all infants received the same standardized complementary diet during the whole first year of life, with the timepoint of introduction to solids as the only difference between groups (Figure 2): Five types of standardized food boxes (Figure 2, Appendix A) with diverse, preprepared complementary foods were available. The boxes followed a step-up concept extending the infants menu according to age and ability to tolerate textures and pieces. A nutritionist calculated and outlined the diet, which was rich in vitamin D, iron, calcium, phosphorus, omega 3 fatty acids, zinc, and folic acid, and offered a varied range of flavors. The study food was commercially available ready-to-use baby jar food, provided for free by Nestle^®^ company (Vienna, Austria). During the study period, parents had to adhere to the diet in more than 80% of the day. To monitor proper adherence to the standardized diet, parents had to complete a self-reported logbook with a food record on three consecutive days, including one weekend day for each study month [13,14].

### 2.5. Primary and Secondary Outcomes

The primary outcome was body height at 12 months, corrected for term. Secondary outcomes were other anthropometric parameters such as weight, head circumference, BMI, and the corresponding z-scores. 

### 2.6. Analysis Sets

The primary data analysis was performed according to the intention-to-treat principle, i.e., all patients who were randomized were included. However, subjects with missing outcome data (lost for follow-up or moved) or withdrawn informed consent were removed from data analysis. In the primary covariate-adjusted analysis, patients with missing covariate values were excluded. In a sensitivity analysis, missing covariate values were multiply imputed. As secondary analysis, a per-protocol analysis was performed, in which patients with less than 80% adherence to the study protocol ascertained by the monthly self-reported feeding protocols were excluded. 

### 2.7. Sample Size Planning

The sample size calculation was based on the data published by Marriott et al. [6], assuming a mean difference in body length of five percent as minimally clinically relevant and further assuming a standard deviation corresponding to a coefficient of variation of 11% and specifying a significance level of 0.05. Under these assumptions, the sample size required to obtain 80% power was calculated for a two-sample *t*-test, which was regarded as conservative approximation to the planned analysis model, resulting in 152 patients (76 for each of the two groups). Initially, an increased sample size to account for up to 30% dropouts was planned, however after an interim analysis in January 2019, the dropout rate was still low, around 11%. Therefore, we adapted the sample size planning from the anticipated 30% dropout rate to a one-to-one replacement for dropouts plus a surplus of five patients. For all data obtained at any time of measurement (e.g., anthropometric data), descriptive statistics were computed for each feeding group. 

### 2.8. Statistical Analysis

For continuous variables, mean and standard deviation were calculated, and graphical descriptive analyses included growth curves and density plots. For ordinal and nominal data obtained at any date of measurement, absolute and relative frequencies were calculated. Dependencies between siblings of multiple birth were not taken into account in the descriptive analysis. For the analysis of the primary endpoint, a linear mixed effects model was fit, explaining body height at visit 5 (12 months of age, corrected for term) through study group (early vs. late introduction of complementary feeding), the baseline body height obtained at visit 1 (at term), nutrition at discharge (three categories breastfed, formula, mixed), gestational age at birth, and sex. A random intercept was included to account for possible correlation between siblings of multiple births. Degrees of freedom were estimated using the Kenward–Roger method. The null hypothesis of no effect due to study group was tested using a *t*-test for the according model coefficient. The covariate-adjusted mean difference in body height between groups and 95% confidence intervals are reported. In a sensitivity analysis, missing covariate values were multiply imputed using multivariate imputation by chained equations, as implemented by the multivariate imputation by chained equations package in R [15]. Missing values for height at visit 1 and nutrition at discharge were imputed using linear regression and polytomous logistic regression, respectively, using weight at visit 1, height at visit 1, nutrition at discharge, sex, and gestational age as predictors. Estimates were pooled from 10 imputed datasets. In a supplementary analysis, the interaction between nutrition at discharge and treatment group was included in the analysis model.

The secondary endpoints weight, height (other than visit 5), BMI, head circumference, and corresponding z-scores were assessed using linear mixed models analogous to the primary analysis, i.e., fixed effects comprised study group, the corresponding baseline value obtained at visit 1, nutrition at discharge, gestational age at birth, and sex. A random intercept was included to account for possible correlation between siblings of multiple births. The covariate-adjusted mean differences in secondary outcomes between groups, and according 95% confidence intervals are reported. The secondary analyses are considered to be of explorative nature and no multiplicity adjustment was applied in the calculation of confidence intervals. SPSS statistical software system (SPSS Inc., Chicago, IL, USA, version 20.0) and the statistical computing software R (R Foundation for Statistical Computing, Vienna, Austria, version 3.5 or higher) were used for all calculations.

## 3. Results

### 3.1. Participants

During a 6.5-year study period between 1 October 2013 and 30 April 2020, 683 infants were screened; 180 met exclusion criteria and 503 were eligible for enrollment in the study. Of these, 326 infants were excluded for the following reasons: parental refusal (*n* = 109), started solids before randomization (*n* = 4), parents wanted to cook for their babies and did not accept preprepared food (*n* = 182), or other reasons, mainly language barriers (*n* = 31). The final cohort included 177 infants: 89 infants in the early and 88 infants in the late group, respectively. All infants received the assigned intervention, but four infants moved or were lost (two infants in each groups) during study period. For four infants (two infants in each group), parental consent was withdrawn, and three infants did not appear at visit 5 (two infants in the early and one infant in the late group) due to acute illness at scheduled study visit; thus, their medical data were not available for primary outcome analysis. Assessment of primary outcome at 12 months, corrected for term, was possible in 83 infants (93%) in the early group and 83 infants (94%) in the late group (Figure 1). Six infants (three infants in each group) did not adhere to the study protocol, hence an additional per protocol analysis was performed. 

### 3.2. Baseline Characteristics, Morbidity and Diet

In the early complementary food group, mean birthweight was 941 (SD ± 253) g and in the late group 932 (SD ± 256; n.s.) g, respectively. Mean gestational age at birth was 27 weeks 1/7 days in both groups. Infant characteristics and outcome data are summarized in Table 1 and were balanced between the two trial groups, except for NEC < grade 3, which was more frequent in the early group (4% versus 0% in the late group). The mean corrected age for term at the timepoint of the introduction of solids was 10.4 (SD ± 0.9) weeks in the early group and 16.2 (SD ± 0.9) weeks in the late group. At discharge, 34% of the infants in the early and 24% in the late group were exclusively breastfeed, 37% in the early and 32% in the late group were formula fed, and 29% in the early and 44% in the late group received mixed feedings. 

### 3.3. Primary Outcome and Secondary Outcomes

Timepoints for scheduled visits and anthropometric measurements were strictly adhered to in both groups (Figure 3). The difference in timepoint between groups at the visit with 12 months, corrected for term, was five days (mean; visit 5 in the early group at 376 (SD ± 31) days and at 371 (SD ± 16) days in the late group; n.s.) At one year, corrected for term, mean height in the early group was 74.7 (SD ± 2.7) cm and 74.4 (SD ± 2.8) cm in the late group (Figure 4). The covariate-adjusted mean difference (early minus late group) was 0.14 cm (95% CI −0.68–0.96, *p* = 0.74). After multiple imputation for missing covariate data in five subjects (four missing height at baseline, one missing type of nutrition at discharge), a covariate-adjusted mean difference of 0.17 cm (95% CI −0.63–0.97, *p* = 0.67) was obtained. In the per protocol analysis, mean height at one year, corrected for gestational age, was 74.4 (SD ± 2.8) cm in the early group and 74.6 (SD ± 2.5) cm in the late group. The covariate-adjusted mean difference per protocol was −0.06 cm (95% CI −0.87–0.76, *p* = 0.89). After imputing missing data for height at baseline in five subjects, the covariate-adjusted mean difference per-protocol was 0.02 cm (95% CI −0.78–0.81, *p* = 0.97). In summary, body height at one year, corrected for term, was not affected by early or late introduction of standardized solids, with confidence intervals ruling out mean differences of more than one cm. This result was robust with respect to different handling of missing data.

There was no difference in any of the other secondary outcome parameters (head circumference, weight, or BMI) except weight z-score at six months, corrected for term (Figure 4, Appendix A). Infants in the early group had significantly higher weight z-scores −0.49 (mean; SD ± 1.2) than infants in the late group −0.56 (mean; SD ± 1.04, *p* = 0.03). This trend persisted until 12 months of age, corrected for term, but did not reach statistical significance. Type of feeding (breast-, mixed-, or formula feeding) did not affect results. The covariate-adjusted differences in these three feeding groups were −0.16 (95% CI −1.69–1.3), 0.14 (95% Cl −1.28 1.5) and 0.38 (95% Cl −9.8–1.73), respectively (Appendix A). 

The red bar represents the time point of introduction of complementary food in the early group (10–12 weeks, corrected for term) and the blue bar represents corresponding time point for the late group (16–18 weeks, corrected for term).

## 4. Discussion

This is the first prospective interventional study in preterm infants investigating the introduction of a standardized complementary diet at two different timepoints and its effect on growth during the first year of life. Early or late introduction of complementary food did not affect height, which is a reliable predictor for skeletal growth and fat-free body mass, but had a transient short-term effect on weight z-scores at six months, indicating a more rapid weight gain in the early complementary food group. Furthermore, this was the first randomized controlled trial conducted in very immature preterm infants with a mean birthweight below 1000 g and a mean gestational age at birth below 28 weeks in both groups.

### 4.1. Growth

In preterm infants, the introduction of complementary food is currently not very well surveyed, and it remains unclear what the right time for weaning from exclusive breast- or formula feeding would be. The challenges are to establish optimal growth, on the one hand, and to consider the delayed neurological abilities that hamper oral food intake, on the other hand. In general, complementary food is introduced early to preterm infants, which was shown in small observational studies [16,17,18,19]. An Italian study documented that infants born between 24–32 weeks were weaned from exclusive nursing or formula feeding at 13 weeks, corrected for term, while more mature infants born between 33–36 weeks were weaned at 15 weeks, corrected for term [4], indicating that the degree of prematurity is a major determinant for complementary food introduction. To date, only two RCTs have investigated time of introduction and nutritional quality of complementary food for preterm infants. The study conducted by Marriott et al. randomized preterm infants either into a “preterm weaning strategy” group (*n* = 37) or into a control group (*n* = 31) [6]. Infants in the preterm weaning strategy group received high-energy, high-protein, semisolid foods together with a preterm infant formula starting at 13 weeks of uncorrected age, provided they had reached at least 3.5 kg body weight. Infants in the control group were started on complementary food at 17 weeks of uncorrected age, provided they weighed at least five kg, and no specific advice for food quality was given. At 12 months of age, infants in the preterm weaning strategy group had faster growth velocity compared to those in the control group (weekly gain in height 5.1 versus 4.9 mm; *p* = 0.04), with no differences in weight or head circumference [6]. Body height is a reliable surrogate for skeletal growth and fat-free body mass, which was also confirmed by body composition measurements [20,21]. However, this study was conducted in the pre-era of post-discharge formula and post-discharge fortification of human milk. At this time, complementary food provided an option for extra nutritional intake to improve nutritional status of preterm infants, while other possibilities were missing. Thus, the results of the study seem to be no longer up to date, and the current findings provide a better understanding of the effects of weaning at different timepoints under contemporary conditions.

Another RCT from India could not find an effect on weight for age z-scores, other anthropometric parameters, or neurodevelopmental outcome at one year in preterm infants with a GA < 34 week starting complementary food at four vs. six months. This is in contrast to the present findings, because we observed a transient effect on weight z-scores at six months, corrected for term, indicating a more rapid weight gain in the early complementary food group after the introduction of solids. However, the Indian study was conducted in a lower–middle-income country [7], indicating that the setting and results cannot be transferred to high-income countries. In this study, infants of both groups showed a remarkable loss in weight for age z-scores of −2.8 around term. This growth retardation persisted up to one year of corrected age, where z-score loss was still −1.6 in both groups. The loss in z-scores does not correspond to normal growth trajectories in European cohorts of preterm infants [22], where the average loss in z-scores at term is −1.2 (mean, SD ± 0.8). In the present study, the average loss in weight z-score at term was −0.63 (mean, SD ± 0.84), which might be related to an effective in-hospital parenteral and enteral nutrition management, and −0.24 (mean, SD ± 1.22) at one year, corrected for term. However, to obtain reliable data of growth and avoid bias, we excluded infants at risk for extrauterine growth retardation due to diseases that might affect stable growth, such as bronchopulmonary dysplasia and necrotizing enterocolitis grade three.

### 4.2. Diet

None of the previous observational studies or RCTs investigated complementary food introduction under standardized conditions using a standardized diet or a feeding protocol. This indicates that comparisons between early and late introduction of solids were not reliable, because it is likely that there was a wide variation in daily food intake between groups, making it impossible to draw a conclusion on nutrient intake and anthropometric outcome. In the present study, both groups had exactly the same diet, which infants and their parents had to adhere to in more than 80% of the study time. The demand to accept the restrictions of standardized diet was a challenge during the recruitment process: one third of the parents denied participation in the study because they wanted to prepare individual meals for their infants, and one fifth of the parents did not want to participate because the diet included ingredients they did not accept for various reasons (e.g., pork for cultural and religious reasons or meat because they wanted a vegetarian diet). Compliance was surveyed by a self-reported food record logbook during the study, where parents had to document daily food and fluid intake. Only three families in each group violated the study protocol and had to be excluded from final analysis.

### 4.3. Considerations for Weaning from Exclusive Breast or Bottle Feeding

The present study showed that the timepoint of introduction of complementary food did not affect growth at one year, corrected for term, and that appropriate growth is not a mandatory reason to start early with complementary food. Considering aspects of an infant’s anatomical, physiological, and oral–motor readiness to tolerate a spoon, the age of 10–12 weeks, corrected for term, seems to be too early to receive oral feedings. In term infants, the earliest gross motor skills, indicative of developmental readiness for spoon-feeding of pureed foods (i.e., holding the head in midline when in supine position and to control its head well when pulled to sitting or at aided sitting), can be observed between three and four months of age [23]. At this age, it can be assumed that the rooting and the extrusion reflexes may have also diminished in some infants. In preterm infants, the necessary developmental milestones for feeding are also reached around the same age range, corrected for term, depending on the severity of illness experienced during the neonatal period, the degree of prematurity, and any sequelae [23]. Although neurological ability to tolerate complementary food was not investigated in this study, it can be assumed that the introduction of complementary food should be driven by the neurological readiness of the infant to tolerate oral feedings instead of considerations of adequate growth. Further research is needed as to whether the timepoint of introduction of complementary food affects iron and vitamin D status, body composition and adiposity parameters, the incidence of nutritional allergies, or neurodevelopmental outcome.

### 4.4. Strengths and Limitations

The main strengths of the study are the standardized conditions in terms of nutritional intake enabling optimal comparability of outcome parameters. By excluding infants with severe diseases that are likely to affect appropriate growth, we improved homogeneity within the study population, which allowed more accurate conclusions on the efficacy of the intervention. In addition, this is the first prospective interventional study on complementary food introduction conducted in the population of very immature preterm infants, and the surveyed data close a long-existing research gap.

A limitation of the study is that due to an imbalance in birthweight after the recruitment status of more than 75%, we had to switch the randomization process to a baseline adaptive randomization design and implemented an additional stratification according to birthweight. A further limitation is that we did not evaluate feeding problems, oromotoric readiness, and neurodevelopmental abilities of the infant, which could have been of interest.

## 5. Conclusions

This prospective interventional study in VLBW infants investigating the impact of an early and a late timepoint for introduction of standardized complementary food on growth did not find a persisting effect on height or anthropometric parameters at one year of age, corrected for term. Results of this study indicate that pediatricians or general practitioners can base their recommendation for introduction of complementary food on the infant’s neurodevelopmental abilities to tolerate oral feedings when parents ask for the optimal timepoint to start weaning the preterm infant from exclusive breast or bottle feeding.

## Figures and Tables

**Figure 1 nutrients-14-00697-f001:**
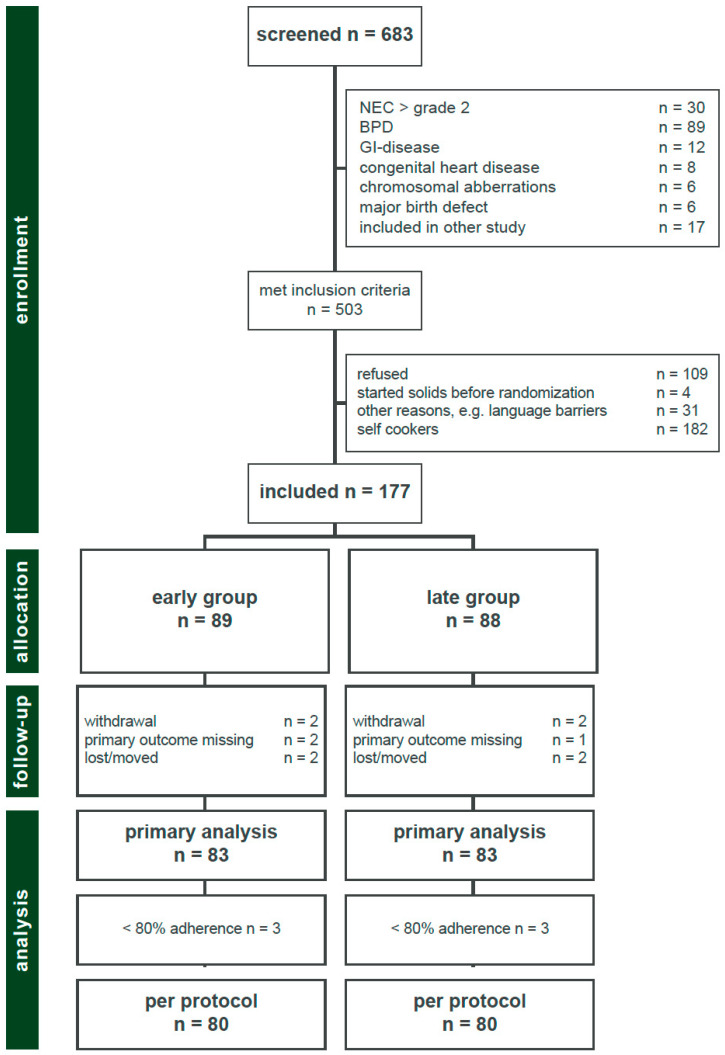
Trial profile.

**Figure 2 nutrients-14-00697-f002:**
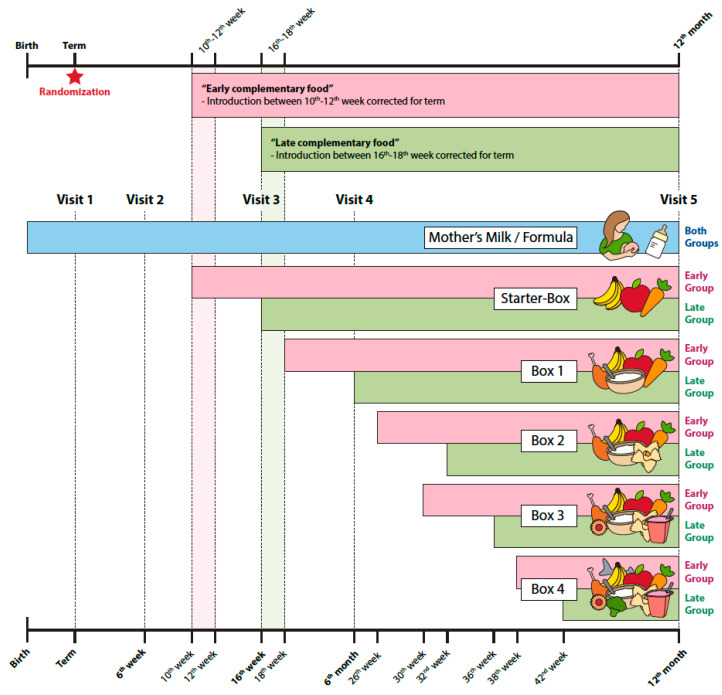
Study flow chart. This figure shows the timeline for randomization, study visits, and the introduction time points of complementary food for early and late study group. The content of feeding boxes is represented by symbols for nutrient groups.

**Figure 3 nutrients-14-00697-f003:**
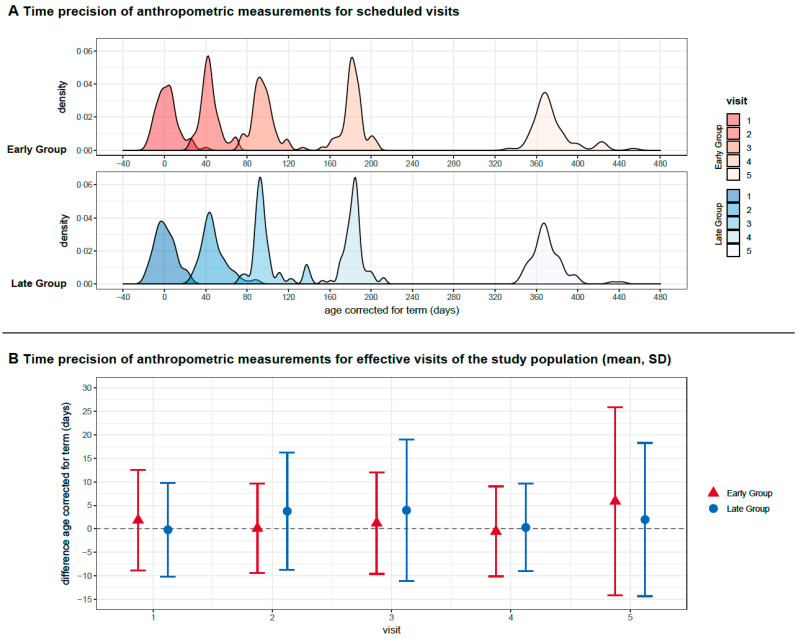
Time precision of anthropometric measurements. Plot A: The Kernel density plot shows the distribution of the time point of anthropometric measurement (*y*-axis) of all study participants for the scheduled visits during the first year of life, corrected for term (*x*-axis). Plot B: Error bars for time point deviation (in days) of effective visits of the study population. Data are given in mean (SD).

**Figure 4 nutrients-14-00697-f004:**
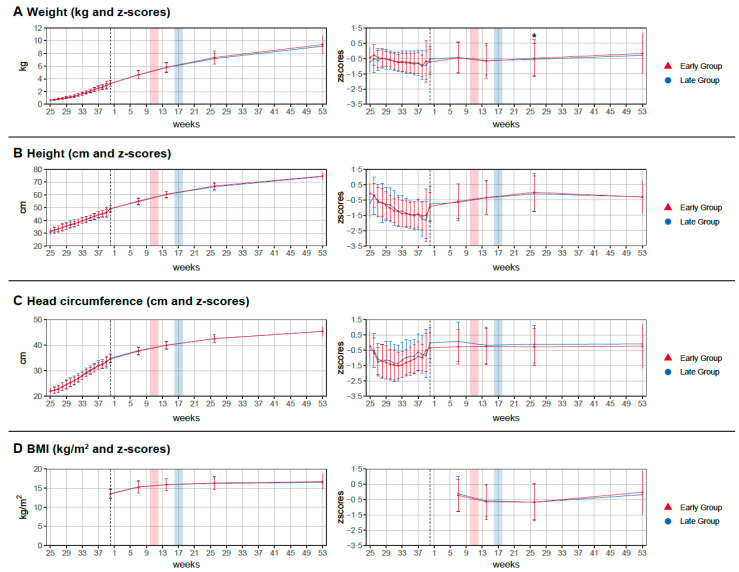
Anthropometric data of the study population. Plot A: This plot shows the weight gain in kilogram and corresponding z-scores of the study population during the first year of life including data from birth to date of expected term. The asterisks mark a significant *p*-value < 0.05. Plot B: This plot shows the height growth in cm and corresponding z-scores of the study population during the first year of life, including data from birth to date of expected term. Plot C: This plot shows the gain in head circumference weight gain in cm and corresponding z-scores of the study population during the first year of life, including data from birth to date of expected term. Plot D: This plot shows the changes in BMI (body mass index) and corresponding z-scores of the study population during the first year of life, including data from birth to date of expected term. Reference values are only available for infants after term; therefore, data before expected date of term are missing.

**Table 1 nutrients-14-00697-t001:** Baseline characteristics of the intention-to-treat population.

	Early Group (*n* = 89)	Late Group (*n* = 88)
Birthweight in g	941 (±253)	932 (±256)
Birth height in cm	34.8 (±3.1)	34.9 (±3.6)
Head circumference at birth in cm	24.8 (±2.2)	24.7 (±2.3)
Gestational age (days) at birth	190 (±16)	190 (±14)
Mean gestational age at birth weeks/days	27/1	27/1
Gestational age (days) at discharge	265 (±12)	265 (±15)
Mean gestational age at discharge weeks/days	37/6	37/6
Male Sex	56 (63%)	42 (48%)
Multiples	32 (36%)	28 (32%)
C-section	78 (88%)	84 (95%)
Antenatal steroids completed	47 (54%)	57 (66%)
Antenatal steroids incomplete	35 (40%)	26 (30%)
SGA	7 (8%)	5 (6%)
NEC I + II	4 (4%)	0 (0%)
PDA	34 (38%)	33 (38%)
ROP Grade 3 and more	5 (6%)	5 (6%)
IVH Grade I-II	9 (10%)	4 (5%)
IVH Grade III-IV	4 (4%)	6 (7%)
PVL	0 (0%)	2 (2%)

Data are given in mean (SD) or *n* (%), SGA = small for gestational age (weight at birth < 10th percentile), NEC = necrotizing enterocolitis, PDA = persisting ductus arteriosus, ROP = retinopathy of prematurity, IVH = intraventricular hemorrhage, PVL = periventricular leucomalacia.

## Data Availability

The study protocol and the individual participant data that underlie the results reported in this article, after de-identification, are available upon request from the corresponding author 6 months after publication. Researchers will need to state the aims of any analyses and provide a methodologically sound proposal. Proposals should be directed to nadja.haiden@meduniwien.ac.at. Data requestors will need to sign a data access agreement and in keeping with patient consent for secondary use, obtain ethical approval for any new analyses.

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
