# Peer review of "Randomized Controlled Trial of Two Timepoints for Introduction of Standardized Complementary Food in Preterm Infants"

_nutrients, 2022, doi:10.3390/nu14030697_

Round 1

Reviewer 1 Report

The manuscript “Randomized controlled trial of two timepoints for introduction of standardized complementary food in preterm infants” has been conducted an interventional study by RCT. First of all, I really appreciate your valuable study for the timing of introduction complementary diet in preterm infants. But there are some major changes to be needed. Please consider all the suggestions carefully and make appropriate alterations to your manuscript.

  1. Your study design excluded preterm infants with high grade NEC, and BPD which you explained the reason in discussion. All neonatal morbidities such as BPD, NEC, PVL, and high grade IVH can affect extrauterine growth in preterm infants. Please explain and describe the reason or evidence in detail why you excluded only NEC stage III, and BPD. And also developmental delay is very important factor which can affect the time for introduction of weaning diet and feeding process, in your study, mean birth weight in participants was <1,000g, which suggests that study groups had risk of development delay in the early life-time. Please explain how you controlled these kinds of confounding factors.
  2. You introduced standardized complementary food in your study. In figure 2, there were content of foods that contain certain nutrients. The volume of food intake can vary individually on a daily basis although participants followed the standardized feeding protocol. The volume of intake can have more important factor on growth in preterm infants than content of food. Please explain how you applied this factor to your results, and how to control or reflect it on process of statistical analysis.

Author Response

point- by- point response:

1. Your study design excluded preterm infants with high grade NEC, and BPD which you explained the reason in discussion. All neonatal morbidities such as BPD, NEC, PVL, and high grade IVH can affect extrauterine growth in preterm infants. Please explain and describe the reason or evidence in detail why you excluded only NEC stage III, and BPD. And also developmental delay is very important factor which can affect the time for introduction of weaning diet and feeding process, in your study, mean birth weight in participants was <1,000g, which suggests that study groups had risk of development delay in the early life-time. Please explain how you controlled these kinds of confounding factors.

Answer by the authors:

We thank the reviewer for this valuable comment. In the present study we wanted to investigate, if the timepoint of introduction of solids affects growth in the first year of life. To get reliable data it was important for us to exclude major factors that might influence our results. In- and exclusion criteria were mainly driven by considerations that distinct morbidities need a special short- and long -term nutritional management and- as a consequence -by concerns that our standardized study diet might affect patients with severe morbidities. Therefore, we excluded infants with diseases that are most frequently and likely to cause growth faltering or growth restriction during the first year of life:  

Infants with NEC III need surgery which is often associated bowel resection, long term parenteral nutrition, intestinal liver failure and short bowel syndrome- depending on the remaining intestine infants need home parenteral nutrition, a special dietetic management with special formula and high caloric intake. Infants with NEC I or II don’t need surgery and bowel resection- usually they recover until discharge or during the first weeks at home. Based on these facts we excluded infants with NEC III and included infants with NEC I and II in our study. We were afraid, that our standardized feeding protocol was not suitable for infants with NEC III and might affect their outcome.

Infants with BPD need a high caloric intake, fluid restriction or other dietetic nutritional management that also couldn’t be covered by our standardized feeding management- therefore this was the second group we excluded from our study. Neurological disabilities caused by IVH or PVL do not necessarily need a special diet in the infants early life- adequate growth depends on their individual development why we didn’t  exclude these infants  from our study.

To show that these morbidities didn’t affect our results we included these them in our statistical model:

Statistical model as described in our paper :

                                                                                   Estimate   low    up          Pr(>|t|)
(Intercept)                                                                     53.50  44.38  62.61     0.00
Group                                                                              0.14 -  0.68     0.96     0.74
Height at visit 1                                                                0.51    0.36    0.67     0.00
GA_in_days                                                                      -0.02 -0.05  0.00     0.10
Sex                                                                                      0.97  0.19  1.74        0.01
factor breastfeeding                                                        -0.29 -1.32  0.73     0.57
factor formula feeding                                                     -0.45 -1.46  0.57     0.38

Statistical model including                                              NEC, IVH and PVL:

(Intercept)                                                                     54.94 45.34 64.54     0.00
Group                                                                                  0.15 -0.69  0.99     0.73
Height at visit 1                                                                    0.51  0.35  0.67     0.00
GA_in_days                                                                       -0.03 -0.06  0.00     0.05
Sex                                                                                          0.97  0.18  1.77     0.02
factor breastfeeding                                                           -0.20 -1.24  0.84     0.70
factor formula feeding                                                        -0.38 -1.41  0.64     0.46
NEC_conservative                                                                 0.99 -1.44  3.42     0.42
IVH_low grade                                                                      -1.20 -2.83  0.43     0.15
IVH_high grade                                                                       0.07 -1.59  1.73     0.94
PVL                                                                                         -0.74 -4.21  2.72     0.67

We also agree with the reviewer that infants with a birthweight below 1000g are at high risk for developmental delay- in the present study neurodevelopmental outcome was/is monitored at 1 and 2 years corrected for term by the Bayley III scales and at 3.5 and 5.5 years by the K-ABC neurodevelopmental test. Yet, we haven’t found a difference in neurodevelopmental outcome at 1 year of age corrected for term as evaluated by the Bayley scales but we ask the reviewer for his kind understanding, that this will be too much information to add in this paper. Data on long and short-term neurodevelopmental outcome will be published in an extra paper.

We also agree that in our study there is a lack of the data about neurodevelopment abilities which is also addressed now more distinct in the strengths and limitations section:

  „A further limitation is that we didn’t evaluate feeding problems, oromotoric readiness and neurodevelopmental abilities of the infant, which could have been of interest.“

2. You introduced standardized complementary food in your study. In figure 2, there were content of foods that contain certain nutrients. The volume of food intake can vary individually on a daily basis although participants followed the standardized feeding protocol. The volume of intake can have more important factor on growth in preterm infants than content of food. Please explain how you applied this factor to your results, and how to control or reflect it on process of statistical analysis.

Answer by the authors:

Thank you for this important comment and we fully agree. To evaluate volume of food intake parents had to complete a self-reported logbook with a standardized food record on three consecutive days including one weekend day once a month during the study period. 3- day food records are a standardized tool in dietetics to monitor food and volume intake (Reference 12+13).

Parents had to document type of food, volume of liquids (e.g. formula, tea and so on) and the exact volume of solids the infant consumed. The exact volume of solid food was determined by weighing the baby jar before and after the meal or if a scale wasn’t available by the number and size of spoons the baby had consumed during one meal. Mean nutritional intake for each makro- and micronutrient and mean caloric intake was calculated for each study month and we couldn’t find significant difference in intake between groups.

However, we found that in general the consumption for e.g. iron and Vitamin D was low but again, we ask the reviewer for his kind understanding that this will be too much information to add in this paper-detailed data on macro- and micronutrient intake of key nutrients such as iron or vitamin D will be published in an extra paper together with laboratory values. 

  1. Toeller, M., et al., Repeatability of three-day dietary records in the EURODIAB IDDM Complications Study. Eur J Clin Nutr, 1997. 51(2): p. 74-80.
  2. Yang, Y.J., et al., Relative validities of 3-day food records and the food frequency questionnaire. Nutr Res Pract, 2010. 4(2): p. 142-8.

Reviewer 2 Report

Haiden et al. performed an interesting RCT, studying on of the most important gap for neonatologists and pediatricians. Despite a lot of study for babies born at term, nowadays there is no consensus regarding the timing of weaning for babies born preterm. I admired the study design and the strict exclusion criteria. To my opinion the study is well-designed and discussed.

However, I have some suggestions for the authors during this revision process.

1) This is not the first study regarding this topic. Boscarino et al. recently published a prospective cohort study on the same topic, with a different methodology. Your results are in line with the this recently published article. Please discuses it and correct "the firts study" in all the text.

https://doi.org/10.3390/children8121085

2) I suggest you to show in all the text (abstract, manuscript and table) mean and SD as mean ± SD, to my opinion is more readable.

3) The graphical aspect of Figure 1 should be improved. Please provide

4) Lines 114-116. I found here and in the other parts of the text that you don’t define the timing for the secondary outcome. Please provide in all the text.

5) Table 1. P value are missing, why? There is no statistical difference between the two study groups for these variables? It appears confused. 4% vs. 0% for NEC is not a so low difference for a statistical difference. Lines 196-198: p values are missing also here. Please provide for all them.

6) As mentioned in the section 4.3, a limitation of the study is the lack of the data about neurodevelopment abilities. This should be added also in limitation section and discussed. Please provide.

7) Finally, why did the authors choose to express the result as BMI? Little agreements are available in the literature for this age for preterm newborns, other indexes could be more suitable (Ponderal Index, Weight for length).

Minor typing errors:

Line 99: “.” in place of “:”

Lines 363-364: please provide to remove.

Author Response

Reviewer 2:

Haiden et al. performed an interesting RCT, studying on of the most important gap for neonatologists and pediatricians. Despite a lot of study for babies born at term, nowadays there is no consensus regarding the timing of weaning for babies born preterm. I admired the study design and the strict exclusion criteria. To my opinion the study is well-designed and discussed.

However, I have some suggestions for the authors during this revision process.

1) This is not the first study regarding this topic. Boscarino et al. recently published a prospective cohort study on the same topic, with a different methodology. Your results are in line with the this recently published article. Please discuses it and correct "the firts study" in all the text.

https://doi.org/10.3390/children8121085

Answer by the authors:

We thank the reviewer for this comment and agree that the paper is on the same topic. However, the present study is a prospective RCT, which is of higher quality than an observational study. Furthermore, the present study was interventional study as we used a standardized feeding regimen, to which the infants had to adhere. Although the studies are on the same topic design and quality are not comparable. In addition, the infants in the cohort study were heavier 1337g (group1) and 1249 g (group 2) and more mature (29 weeks of gestation at birth). So the present study remains the first interventional trial in very immature preterm infants with a mean birthweight below 1000gram and a mean gestational age at birth below 28 weeks.

However, we changed the wording according to the reviewers suggestion and hope that this rewording is acceptable for the reviewer:

“Furthermore, this was the first randomized controlled trial conducted in very immature preterm infants with a mean birthweight below 1000 gram and a mean gestational age at birth below 28 weeks in both groups.”

In addition, we added the study to the other observational studies and cited the study in the discussion. The new reference was inserted as reference 18:

Boscarino, G., et al., Complementary Feeding and Growth in Infants Born Preterm: A 12 Months Follow-Up Study. Children (Basel), 2021. 8(12).

2) I suggest you to show in all the text (abstract, manuscript and table) mean and SD as mean ± SD, to my opinion is more readable.

Answer by the reviewers. The values were replaced according to the reviewers helpful suggestion.

3) The graphical aspect of Figure 1 should be improved. Please provide

Answer by the authors:

We thank the reviewer for his/her feedback and redesigned Figure 1

4) Lines 114-116. I found here and in the other parts of the text that you don’t define the timing for the secondary outcome. Please provide in all the text.

Answer by the authors:

We thank the authors for this valuable comment. We completed the sentence in the section primary and secondary outcomes:

Secondary outcomes were other anthropometric parameters such as weight, head circumference, BMI, and the corresponding z-scores at 12 months corrected for term.

5) Table 1. P value are missing, why? There is no statistical difference between the two study groups for these variables? It appears confused. 4% vs. 0% for NEC is not a so low difference for a statistical difference.

Answer by the authors:

We thank the authors for their valuable reply- most of the top journals don’t request p- values for baseline characteristics- that's why we skipped them

According to the CONSORT statement, statistical testing of baseline characteristics is not recommended for randomized trials, as observed differences in baseline characteristics (e.g., NEC rates) are the results of chance, not bias (http://www.consort-statement.org/checklists/view/657-harms/1032-baseline-data). This sort of hypothesis testing may therefore mislead investigators and readers, wherefore we did not include them in our baseline characteristics table. However, we are happy to be guided by the editors’ views.

Nevertheless, we included p-values for the reviewer in this point-to-point reply (Table 1b).

Regarding the results of the statistical hypothesis testing of our baseline characteristics the respected reviewer suggested, the resultant distribution for NEC is the result of chance, not bias, due to appropriate randomization procedure of our study.

Early Group (n=89)

Late Group (n=88)

p- value

Birthweight in g

941 (±253)

932 (±256)

0.82

Birthheight in cm

34.8 (±3.1)

34.9 (±3.6)

0.82

Head Circumference at birth in cm

24.8 (±2.2)

24.7 (±2.3)

0.91

Gestational age (days) at birth

190 (±16)

190 (±14)

0.95

Mean Gestational age at birth weeks/days

27/1

27/1

Gestational age (days) at discharge

265 (±12)

265 (±15)

0.95

Mean Gestational age at discharge weeks/days

37/6

37/6

Male Sex

56 (63%)

42 (48%)

0.06

Multiples

32 (36%)

28 (32%)

0.67

C-section

78 (88%)

84 (95%)

0.11

Antenatal steroids completed

47 (54%)

57 (66%)

0.16

Antenatal steroids incomplete

35 (40%)

26 (30%)

0.2

SGA

7 (8%)

5 (6%)

0.78

NEC I+II

4 (4%)

0 (0%)

0.13

PDA

34 (38%)

33 (38%)

1

ROP Grade 3 and more

5 (6%)

5 (6%)

1

IVH Grade I-II

9 (10%)

4 (5%)

0.73

IVH Grade III-IV

4 (4%)

6 (7%)

0.26

PVL

0 (0%)

2 (2%)

0.47

Table 1b: Baseline characteristics of the intention-to-treat population

Lines 196-198: p values are missing also here. Please provide for all them

Answer by the authors :

This is the text from line 196-198:

During a 6·5-year study period between first October 2013 and 30th April 2020, 683 infants were screened, 180 met exclusion criteria and 503 were eligible for enrollment in the study. Of these, 326 infants were excluded for the following reasons: parental refusal (n=109), started solids before randomization (n=4), parents wanted to cook for their babies and did not accept pre-prepared food (n=182) or other reasons, mainly language barriers (n=31)

We are sorry but we can’t provide p-values for subjects before they were randomized into groups. After randomization there was no difference between groups.

6) As mentioned in the section 4.3, a limitation of the study is the lack of the data about neurodevelopment abilities. This should be added also in limitation section and discussed. Please provide.

Answer by the reviewers:

We thank the reviewer for this helpful statement and inserted the following text in the strengths and limitation section:

„A further limitation is that we didn’t evaluate feeding problems, oromotoric readiness and neurodevelopmental abilities of the infant, which could have been of interest.“

7) Finally, why did the authors choose to express the result as BMI? Little agreements are available in the literature for this age for preterm newborns, other indexes could be more suitable (Ponderal Index, Weight for length).

Answer by the authors:

As suggested by the respected reviewer, we added additional indices of body proportionality (Ponderal Index and weight-for-length z-scores). As in BMI z-scores, in both additional indices no significant differences were identified between the early and late feeding group. Weight-for-length z-scores display a similar longitudinal decrement with a nadir at from 6 weeks corrected for age to 6 months corrected for age, and a similar increment until 12 months corrected for age in both groups respectively. Upon longitudinal observation the Ponderal Index displays a slight increment in both groups decreasing to its baseline value, as calculated at birth, at 12 months corrected for age. However, we did not compute statistical models for the longitudinal interpretation of this post-hoc analysis as this was none of our a-priori set endpoints, wherefore further interpretation of this descriptive observation is warranted. Regarding body proportionality indices a previously published study by Ferguson et al. (https://doi.org/10.1159/000480118) in two large datasets of preterm infants (original dataset n=130,111, validation dataset n=1127,744) suggests that BMI is the best measurement of body proportionality for preterm infants, as the Ponderal Index overcorrects and weight-for-length undercorrects for length. However, deduction to our results is limited, as the study by Ferguson et al. solely includes preterm infants during their NICU stay, and no measurements after discharge. Nevertheless, the current body of literature suggests that clarity on body proportionality indices for preterm infants is still not achieved, advocating reporting of different body proportionality measurements in studies on preterm infants as well as conducting further studies on body proportionality in terms of body composition, especially by utilizing gold-standard methods such as whole-body-densitometry by e.g., air displacement plethysmography.

Minor typing errors:

Line 99: “.” in place of “:”

Lines 363-364: please provide to remove.

Answer by the authors:

The minor typing errors were corrected according to the reviewers helpful suggestion

Round 2

Reviewer 1 Report

Thank you for your effort to revise your article what I mentioned. 

Reviewer 2 Report

Thank you for the clear answers. To my opinion the manuscript appears improved and ready for the publication.

Congratulations!